# The Possibility of Reduction of Synthetic Preservative E 250 in Canned Pork

**DOI:** 10.3390/foods9121869

**Published:** 2020-12-15

**Authors:** Karolina Ferysiuk, Karolina M. Wójciak

**Affiliations:** Department of Animal Raw Materials Technology, Faculty of Food Science and Biotechnology, University of Life Sciences in Lublin, Skromna 8 Street, 20-704 Lublin, Poland; karolina.ferysiuk@student.up.edu.pl

**Keywords:** antioxidant activity, nitrite-reduced meat product, color parameter, PCA, hazards

## Abstract

The purpose of this study was to determine the possibility of reducing the amount of NaNO_2_ added to canned pork during 180 days of storage. In this study, three variants of canned pork were prepared by adding different amounts of sodium nitrite: N (100 mg/kg), NH (50 mg/kg), and NF (no nitrite). The antioxidant capacity, amount of secondary products of lipid oxidation, color intensity, and pH were analyzed after one, 60, 90, and 180 days of storage where sensory properties, water activity (a_w_), selected pathogenic bacteria, nitrate and nitrite residues, *N*-nitrosamines (NA), and cholesterol were analyzed after 1 and 180 days of storage. The redness parameter of the nitrite-free canned meat was found to be significantly lower (about 6.4) than that of the products containing sodium nitrite (N: 10.49 and NH: 9.89). During the storage period *C. perfringens*, *L. monocytogenes*, and *Salmonella* were detected in the products. It is not possible to completely eliminate nitrite from the canned pork production process without deteriorating the color, antioxidant properties, sensory characteristics, and health safety. However, the level of hazard chemicals such as NA, nitrate and nitrite residues can be limited by decreasing the amount of nitrite addition to 50 mg/kg. The free-radical scavenging ability for the sample with 50 mg/kg of sodium nitrite was observed to be poor, so its fortification with plant material rich in various polyphenolic substances may be necessary.

## 1. Introduction

Nitrite has been used as a meat additive since ancient times, and the relationship between the addition of salt and the characteristic red color of the cured meat was probably first noticed during the Roman period. However, the discovery and complete understanding of the curing mechanism was achieved only at the end of the 19th century [1] (pp. 85–87). The most characteristic feature of nitrite is its ability to create a reddish-pink color on meat products through complicated reaction steps that result in the formation of nitric oxide (NO). NO reacts with a heme protein called myoglobin, and the underlying reaction becomes stabilized during thermal treatment [2,3,4]. In addition, nitrite exerts a positive effect on the meat products, making them safe for consumption and extending their shelf-life. Alahakoon et al. [3] pointed out that, due to its ability to limit the activity of the prooxidant heme iron, nitrite can slow down the lipid oxidation process and the development of rancidity. The authors also indicated that nitrite possesses strong microbial inhibitory properties and prevents the growth of pathogenic bacteria such as *Clostridium botulinum, Clostridium perfringens, Listeria monocytogenes,* and *Bacillus cereus*. Furthermore, nitrite imparts a characteristic flavor and taste to cured meat products [3].

Unfortunately, a discovery the carcinogenic N-nitrosamines (NA) in fried bacon made in the 1971 raised concerns about the safety of nitrite use which remains unanswered to date [5]. This led to a wide interest in the formation of NA in meat products and their influence on human health. NA can be generated during the usual processes applied to the products at home (e.g., cooking, frying), or in the products formed during the production process, or in the gastrointestinal environment through endogenous reactions. A common factor involved in the generation of NA is the reaction between a secondary amine and a nitrosating agent [4,6,7]. Moreover, a very high intake of nitrite can also lead to methemoglobinemia, a condition in which nitrite binds to hemoglobin and impairs the oxygen transport to cells [8]. The current directive 2006/52/EC [9] and the Regulation No. 1333/2008 [10] state that the maximum amount of nitrite that may be added to the sterilized meat products with F_0_ > 3.0 (an F_0_ value of 3 is equivalent to 3 min of heating at 121 °C which results in the reduction of the bacterial load from one billion spores to one spore in a thousand cans) is 100 mg/kg.

Therefore, researchers focus on investigating if it would be possible to reduce nitrite for meat products [6]. However, this change in the amount of added nitrite will vary for the different categories of the products and will strongly depend on the recommendations for the microbiological safety as well as other factors such as storage temperature, water activity, and pH value [6]. The possibility of reduction of nitrite addition is analyzed by scientists for fermented meat product [11,12,13] and cooked meat product [14]. As it could be noted, a problem with nitrite addition to meat products still exists and various methods of replacement of this substance are still considered [15]. A very popular method is to reduce the amount of nitrite together with plane extract addition. However, in some cases, this can lead to an improvement of some properties and the deterioration of others [16,17,18,19]. However, there is a lack of information about the possibility of reducing the amount of nitrite added to canned meat in the available literature [20].

The goal of the present work is to determine the lowest possible amount of nitrite that can be added to canned meat in comparison to the recommended levels [10] in order to ensure the highest level of protection of human health against exposure to microbiological, chemical, and physical hazards. The potential microbiological and chemical hazardous compounds affected by the addition of sodium nitrate, their mechanisms of generation, their associated health risks, and ways of preventing their formation in meat products are briefly described in Table 1.

Therefore, the objective of the present work was to determine the possibility of reducing the amount of sodium nitrite (III) added to canned pork taking into consideration the health safety, antioxidant stability, and sensory characteristics during 180 days of storage (4 °C). Canned meat products were subjected to close examination, and their color, antioxidant properties, sensory characteristics, and health safety, including the content of secondary oxidation products, NA, and nitrite and nitrate residues, as well as the presence of selected pathogenic bacteria, were analyzed.

## 2. Materials and Methods

### 2.1. Canned Pork Preparation

The materials used for preparing canned pork were pork dewlap and pork shoulder obtained from an organic farm (Zakład Mięsny Wasąg SP. J., Poland, organic certificate no: PL-EKO-093027/18). The obtained samples were subjected to initial grinding and divided into three variants, to which different amounts of sodium nitrite were added: N, 100 mg/kg; NH, 50 mg/kg; and NF, no nitrite. After 24 h of curing process the meat samples were ground again using a universal machine type KU2-3E (Mesko-AGD, Skarżysko-Kamienna, Poland; diameter of the mesh—5 mm). The recipe for each variant was as follows: 20% of pork dewlap, 80% of pork shoulder, 5% of technological water, and 2% of salt. After the mixing process (KU2-3E, Mesko-AGD, Skarżysko-Kamienna, Poland; 4–5 min of mixing for each variant), the samples were put into cans until they were almost full (180 g), closed tightly, and subjected to thermal treatment in a sterilizer (vertical steam sterilizer, TYP-AS2, SMS, Warszawa, Poland). The experimental canned pork was heated at 121 °C, assuming that their degree of heating was achieved measured with the sterilization value will be F ≈ 4 min. The sterilization value of F was determined from the Equations (1) and (2):(1)$$F=\int_{0}^{1}\mathrm{Ldt}$$
(2)$$L=10^{\frac{T-T_{0}}{z}}$$
where F is the sterilization value, L the lethality degree, T_0_ the reference temperature (121 °C), and z the sterilization effect factor (10 °C).

The sterilization values were determined by determining the degree lethality by measuring temperature every minute. The limits of integration were assumed from 90 °C during the growth phase and to 90 °C during the decrease, i.e., cooling. The degree of heating of the cans was determined in their critical zone by means of an electric thermometer equipped with a thermoelectric sensor.

After sterilization, the products were cooled in water and stored in the refrigerator at 4 °C for 180 days. The products were divided into four groups, and each group was tested immediately after production (day 0/ 1st day), and after 60, 90, and 180 days of storage. The experiment included one-time preparation of 36 (+5 inventory) canned pork (12 cans from each test variant: N, NH, NF). In each study period (1, 60, 90, 180 days) 3 cans of each variant were tested. The experiment thus planned was repeated three times in about two weeks’ time interval. Considering the three research series, a total of at least 108 cans have been produced.

### 2.2. Quality Attributes

#### 2.2.1. Color Attributes and Nitrosylhemochrome Content

The concentration of nitrosylhemochrome (Mb-NO) was measured in the products according to the method described by Hornsey [27]. Homogenized with acid acetone (40 mL) (P.P.H. “STANLAB” Sp. J.) minced samples (5 g) has been stored in dark conditions for 30 min. (room temperature) before could be centrifuged and tested. This was done by multiplying the absorbance measured at 540 nm by 290. The resulting value was expressed as mg/kg.

The color of canned pork was determined using the CIE L*a*b* scale [28], and the results were expressed as L* (lightness), a* (redness), and b* (yellowness). All these color parameters were measured using an X-Rite Color 8200 spectrophotometer (X-Rite Inc., Grand Rapids, MI, USA, the size of the port—13 mm, standard observer—10°, illuminant D65). Immediately after opening the can, the meat-fat block was cleaned of aspic and then cuboid of 3 × 5.5 [cm] was cut to measure the color. Each process took 10 min. The color was always measured 10 min. after opening the can in order to stabilize it.

#### 2.2.2. Antioxidant Capacity

The DPPH^·^ (2,2-Diphenyl-1-picrylhydrazyl) and ABTS^·+^ (2,2′-azinobis-(-3ethylbenzothiazoline-6-sulfononic acid) (Sigma-Aldrich) assays were carried out based on the method described by Jung et al. [29] with some modifications. To obtain a supernatant, 5 g of the minced sample was homogenized with 10 mL of ethanol. Next samples were centrifuged and filtered. The ABTS·+ solution was diluted with ethanol to obtain a absorbance 0.7 ± 0.02 at 734 nm and next, to conduct the experiment, to 1.8 mL of the ABTS^·+^ solution 12 μL of supernatant was added. The DPPH· solutions were diluted with ethanol until 0.9 ± 0.02 absorbance was obtained at 517 nm. In both cases, the absorbance of the solutions was measured after 3 min using a U-5100 UV-VIS spectrophotometer (HITACHI High America Inc., Schaumburg, IL, USA). The DPPH^·^ radical scavenging and ABTS^·+^ reducing capacity were calculated from the standard curve of Trolox equivalent (concentration 15 mg/mL to 0 mg/mg for ABTS, and 0.025 mg/mL to 0 mg/mL for DPPH) (Acros Organics™, New Jersey, USA), and the results were expressed as mg Trolox/mL. In addition, the ferric ions (Fe^3+^) reducing antioxidant power assay (FRAP) was performed, using the method described by Oyaizu [30], and the obtained results were expressed as absorbance measured at 700 nm.

#### 2.2.3. Secondary Lipid Oxidation Products

The content of secondary products of lipid oxidation was measured according to the method described by Fan et al. [31], where tested, minced sample (5 g) has been homogenized with mixture of trichloroacetic acid (TCA–7.5%) (POCH S.A.) with ethylenediaminetetraacetic acid (EDTA) (Sigma-Aldrich) (0.1%). Next, samples were shaken, filtered, and heated in 100 °C after the 2-thiobarbituric acid (TBA) (POCH S.A.) addition. After cooling and centrifugation, the TBARS content were measured using U-5100 UV-VIS spectrophotometer (HITACHI High America Inc., Schaumburg, IL, USA). The level of TBARS was estimated using the following Equation (3), and the result was expressed as malondialdehyde (MDA) mg/kg:TBARS = (A_532_ – A_600_)/(155·1/10·72,6) ·1000(3)
where “A” refers to the absorbance measured at 532 and 600 nm.

#### 2.2.4. pH Value

The pH value of the products was measured using a CPC-501 pH meter (Elmetron, Poland) which was fitted with an ERH-111 pH electrode (Hydromet, Gliwice, Poland). For determining the pH value, minced caned pork (2.5 g) were homogenized with 25 mL of distilled water using a homogenizer (IKA® T25, IKA-Werke GmbH & CO. KG, Staufen, Germany).

#### 2.2.5. N-nitrosamines

The volatile N-nitrosamines were analyzed according to the method of Drabik Markiewicz et al. [32] and DeMey et al. [33]. Meat samples (50 g) were mixed with 200 mL 3 N potassium hydroxide (KOH). The volatile N-nitrosamines were extracted from the meat samples by vacuum distillation bridge with two roundbottom flask. After addition of 4 mL 37% hydrochloric acid (HCl), the distillate was extracted three times with 50 mL of dichloromethane. Subsequently, the extract was concentrated using Turbo Vap®. For the detection and quantification of selected N-nitrosamines (NDBA—N-nitrosodibutylamine, NDMA—N-nitrosodimethylamine, NDEA—N- nitrosodiethylamine, NDBA—N-nitrosodibutylamine, NMOR—N-nitrosomorpholine, NPIP—N-nitrosopiperidine, NPYR—N-nitrosopyrrolidine), a gas chromatograph coupled to a thermal energy analyzer (z. B. GC 7890B /Thermal Energy Analyser z. B. Tea 810 von der Fa. Ellutia) was used. The extracts (5 μL) were injected in/into a packed column and a chromatographic separation was carried out by using argon as a carrier gas (25 mL/min). The injection port of the GC went in a ramp from 155‒255 °C and the oven temperature was increased from 60 °C to 250 °C at 5 °C/min. The temperature of the pyrolizer of the TEA was set at 500 °C. The content of N-nitrosamines was estimated after 1 and 180 days of storage.

#### 2.2.6. Cholesterol Content

Cholesterol was extracted, by direct saponification, according to the method described by Bandeira et al. [34] with some modification. The digested samples (2 g) were alkaline-saponified with 4 mL of a 50% aqueous solution of potassium hydroxide (KOH), and the unsaponified cholesterol is extracted with 10 mL of petroleum ether. In this study, following the enzymatic digestion, saponification, and extraction procedures, the solvent was distilled off and the residue was dissolved in n-heptane. The analysis was performed using gas chromatograph equipped with flame ionization detector (GC-FID). The contents were calculated against a calibration series and an internal standard, taking into account the volumes of petroleum benzine used. The limit of determination was 10 mg/kg. For the analysis of the samples, gas chromatograph GC, model GC-2010 Shimadzu (Tokyo, Japan) was used with a flame ionization detector (FID), 0.25 µm analytical column of dimension 30 m × 0.25 mm containing a polyethylene VA-WAX (CA, USA) stationary phase, and appropriate software (GC Solution). Hydrogen was used as the carrier gas, while nitrogen was employed as the make-up gas. Compressed air was used as the carburant. The ratio of the flow rates of N2, H2, and synthetic air was 30:40:400 mL/min. The gas chromatographic conditions used in this study were set as follows: split injection: 20:1, injection volume: 1.0 µL, and temperature: 250 °C. The FID temperature was 320 °C.

#### 2.2.7. Water Activity

The water activity (aw) parameter was assessed in the homogenized products at 25 °C using a LabMaster-aw analyzer (Novasina AG, Lachen, Switzerland). Samples of minced meat were applied to vessels dedicated for testing water activity (up to half the height of the vessels).

#### 2.2.8. Nitrate and Nitrite Residues

The content of nitrate and nitrite residues was measured in the product based on EN 12014‒4:2005 [35] using FIAstarTM 5000 Analyzer (Foos, Denmark) equipment. 10 g of meat sample was homogenized in water and clarified with acetonitrile before taking the measurement by ion-exchange chromatography (IC) and ultraviolet (UV) detection at a wavelength of 205 nm. The content of nitrate and nitrite residues was determined in the products after 1 and 180 days of storage and the residual nitrite and nitrate amount was given as mg/kg NaNO2 and NaNO3 respectively.

#### 2.2.9. Microbiological Evaluation

Microbiological evaluation was performed in the products in order to check for the presence of selected types of pathogenic bacteria. Twenty grams of the sample were homogenized with 180 mL of peptone water for 1 min in a Stomacher Lab- Blender 400 (Seward Medical, London, UK) and decimal dilutions were prepared. The total aerobic mesophilic bacteria count was determined by Plate Count Agar (Merck Co, Darmstadt, Germany) and incubated at 30 °C for 72 h. The population of Clostridium perfringens. was determined by spread plating counting into TSC agar anaerobically (Merck Co, Darmstadt, Germany) and incubated at 37 °C for 20 ± 2 h. The results are reported as colony forming units per gram of product (cfu/g). The analysis was carried out in accordance with the following Standards: *C. perfringens* [36], *Listeria monocytogenes* [37], and *Salmonella* [38]. The total aerobic mesophilic bacteria (TVC) were measured according with PN-EN ISO 4833-1:2013-12 [39]. The products were subjected to evaluation after 1 and 180 days of storage.

#### 2.2.10. Sensory Analysis

The sensory characteristics of canned meat products were assessed according to the method described in ISO/DIS 13299.2:1998 [40]. This analysis was carried out by scientific staff (ten members) from the Department of Animals Raw Materials Technology at the University of Life Sciences in Lublin, Poland. Briefly, canned meat products were cut into equal slices and presented to the panelists. First of all, based on expert discussions, a product evaluation card was developed with a list of distinguishing features and accepted minima and maxima to determine the intensity of the trait. Then there were two sessions in which each expert individually assessed the products using the prepared assessment card. Samples of each product were blinded, served in containers with a unique sample number in three series. The tastings took place in two sessions. Determination of the intensity of the features was carried out using an unstructured scale, anchored at an appropriate distance from the edges of the score card with words specifying the minimum and maximum intensity of the examined feature. A linear (100-mm line which is converted into conventional units (c.u.)) graphic scale was used for the assessment of sensory properties. The parameters analyzed and the descriptors used for each of them were as follows: meat color—from gray to pink, fat color—from yellow to white, compactness—from low to high, juiciness—from dry to juicy, and hardness and overall quality—from low to very high. For color (green), taste (intensity of rancid taste, sour taste, salty taste, bitter taste, spicy taste, metallic taste, herbal taste, or other taste), and aroma (intensity of rancid aroma, acid aroma, spicy aroma, metallic aroma, herbal aroma, or other aromas), descriptors ranging from none to very high intensity were adopted. The analysis of sensory properties was carried out after 1st and 180 days of storage. During this all time, samples were storage in the refrigerator (4 °C) (this also applies to samples already removed from the package, cut, and prepared for sensory analysis that day. However, prepared samples were removed from the refrigerator before the arrival of the tester for the samples stabilization).

### 2.3. Statistical Analysis

The two-way analysis-of-variance model included the main effects of the level of nitrite (100 mg/kg, 50 mg/kg, none) and the storage period (1, 60, 90, and 180 days) as well as their interactions was used. For sensory analysis applied linear mixed model (y = beta0 + beta1X + u1Z1 + u2Z2 + epsilon; where y—analyzed variable (meat color, fat color etc.), X—belonging to a group (N, NF, NH), Z1—belonging to a specific sample, Z2—belonging to a specific session. X was a fixed effect, so beta1 was treated as a fixed number and was estimated. The Z1 and Z2 were a random effect, so u1 and u2 were treated as a random variable and were not estimated. The basic statistical and multivariate analyses (principal component analysis [PCA]) were performed using the statistical package R, version 3.6.1. The significance of the differences between mean values was calculated using Tukey’s range t-test at a level of *p* < 0.05. The data matrix was autoscaled before applying the chemometric method.

## 3. Results and Discussion

### 3.1. Color Attributes and Nitrosylhemochrome Content Analysis

The most important factor that influences the consumer’s decision about purchasing meat and meat products is color, which creates the first impression about a product as well as indicates its freshness [41,42]. The effects of various amounts of nitrite addition on the color attributes and nitrosylhemochrome content of canned meat are presented in Table 2. For lightness (L*) parameter, the only significant (*p* < 0.001) differences were observed directly after production and after 90 days of storage, and samples with nitrite addition (N and NH) presented lower L* values (61.37 mg/kg and 62.28 mg/kg—day 1, 62.95 and 62.92—day 90, respectively) than the nitrite-free sample (63.72 and 65.73 mg/kg). The lower L* values of the N and NH samples could appear due to the direct addition of nitrite. Storage time significantly affected the L*parameter. In all research variants, a significant increase (*p* < 0.05) in the L* parameter was noted after 180 days of storage indicating that the meat color was systematically fading during storage. The breakdown of heme pigments during storage might have led to the lower lightness value [43]. This phenomenon was especially visible in the NH and NF tests, where the residual nitrite was lower than 10 mg/kg (Table 5). According to Sindelar and Milkowski [44] a residual nitrite level of 10–15 mg/kg is recommended as a reservoir primarily for the regeneration of cured meat color as it counteracts fading over the storage period.

Based on the measured a* and b* values, the NF sample was characterized as the least red and the most yellow sample. Furthermore, reducing the amount of nitrite by half did not adversely affect the tested color parameters. The results of the color analysis showed that the NH sample (with 50 mg/kg of nitrite) achieved the same or similar values of a* and b* parameters as the N sample (with 100 mg/kg of nitrite) during the entire storage period.

The exception was the study carried out at the end of the storage period, in which in the NH significantly lower (*p* < 0.05) value of the a* parameter was recorded compared to the N sample. This finding was also confirmed by the sensory analysis (Table 7) in which the color of the meat (redness) was observed to be significantly higher in the N and NH samples compared to the NF sample, as shown in Figure 1. Most likely, too low content of residual nitrites in canned pork after 180 days of storage (below 10 mg/kg) caused a decrease in the share of red in the overall color tone [44]. Eskandari et al. [45] noted the decrease of redness in frankfurters with 40 mg/kg of nitrite addition. Following Alahakoon et al. [3] to achieve the desired color on cured meat, a minimum of (2–14 mg/kg) of nitrite should be added, while Rivera et al. [4] stated that the minimum amount added should be (10–15 mg/kg). However, according to the report created by FCEC [6], the amount of nitrite added to the nontraditional meat products for achieving color formation and avoid non-uniform curing, should range between 55 and 70 mg/kg. Moreover, the scientific panel pointed out that for ensuring the stability of color during storage, the amount of nitrite added should be about 80 mg/kg. In our study, nitrite-free sample presented significantly lower (*p* < 0.001) value of this parameter and the differences between two samples with nitrite addition have occurred only at the end of the storage period.

As expected, the amount of nitrosylhemochrome (NO‒Mb) was found to be significantly (*p* < 0.001) lower in nitrite-free sample (NF = 1.16–3.31 mg/kg) compared to the samples containing sodium (III) nitrite (NH = 18.37–26.52 mg/kg and N = 28.16–32.27 mg/kg) at all the analyzed time points (Table 2). Thus, nitrite addition significantly influenced the amount of NO-Mb in samples after 60, 90, and 180 days of storage. Very low amount of nitrosylhemochrome observed in the NF sample can be the result of the lack of nitrite (Table 2). In the study on uncured meat products, it was reported that thermal processing caused the denaturation of myoglobin to ferrihemochrome resulting in the formation of a dull-brown color [46]. This observation agrees with the results of our study for nitrite-free sample (Table 2, Figure 1). According to the results of our study, reduction in the amount of nitrite, added to canned meat (50 mg/kg) caused a severe decrease in both the color parameter (L*a*b*) and the level of nitrosylhemochrome during storage. The color of the NH sample was brighter and less red compared to the control N (Table 2, Figure 1).

### 3.2. Antioxidant Capacity Analysis

The antioxidant capacity of canned pork was measured through ABTS^·+^ reducing activity, DPPH radical scavenging activity, and ferric ions (Fe^3+^) reducing antioxidant power assay (FRAP) (Table 3). The results obtained through DPPH^·^ and ABTS^·+^ methods for samples containing various amounts of nitrite were expressed as mg Trolox/mL.

Usually, at least two methods are applied in order to investigate various mechanisms related to antioxidant properties [47]. The most common methods are ABTS^·+^ and DPPH—both related to mechanism of free radical inhibition [48]. In contrast to the DPPH method, ABTS^·+^ is soluble not only in organic but also in aqueous media, and therefore lipophilic and hydrophilic nature of various compounds can be measured [48,49]. The FRAP method is connected to the reduction of iron ion from form Fe^3+^ (ferric state) to Fe^2+^ (ferrous state) which is not only connected with the color of meat products but also with its durability [50,51].

At the beginning of the storage, sample N characterized by the highest antioxidant capacity measured by ABTS^·+^ method—5.62 mg Trolox/mL was compared to the samples NH and NF (5.09 mg Trolox/mL and 4.79 mg Trolox/mL respectively). With time, the antioxidant properties have been reduced for all samples. However, the strongest reduction has been noted for sample with 100 mg/kg nitrite addition after 180 days—change for about 3.5 mg/mL (for sample with reduced amount of nitrite this change was 2.26 mg/mL). Although, it was noted that after 60 and 90 days of storage no significant differences (*p* < 0.001) have been noted for samples with nitrite addition. In case of DPPH parameters, alternating increases and decreases in value of NF sample has been noted. Canned meat with reduced amount of nitrite addition presented gradual decrease of value (at day 1 from 0.017 mg/mL to 0.014 mg/mL at day 180) in contrast to sample N which presented stable antioxidant properties during time. However, the sample with no nitrite addition presented better antioxidant properties than other samples (0.019 mg/mL at the end of the storage).

Good antioxidant properties presented by nitrite-free sample have been observed by other authors. Seo et al. [52] noted that sausage with no nitrite addition presented higher antioxidant capacity (DPPH) than sausage with nitrite addition (70 mg/kg) after 1st day of storage. The situation began to change over time and finally after 30 days of refrigerated storage the antioxidant capacity significantly decreased. Also, Karwowska et al. [53] noted that total antioxidant properties of cooked meat extracts with various amount of sodium nitrite addition (0 mg/kg, 50 mg/kg, 100 mg/kg, 150 mg/kg) were higher for nitrite-free sample than for samples with nitrite addition. This situation is explained by the activity of peptides arising from protein muscle. Although, analysis of protein antioxidant capacity showed that samples with nitrite addition presented stronger antioxidant properties against ABTS˙^+^ radical than DPPH. In context of DPPH radical, samples presented similar activity.

After 1st day of storage, the sample with 100 mg per kg of sodium nitrite addition presented higher iron reducing capacity (FRAP method) than sample with reduced amount of nitrite (1.99 and 1.5 respectively). The situation changed after two months of storage. Sample NH presented stronger reducing properties than sample N. At the end of storage, both samples with nitrite addition achieved similar iron reducing capacity. Fe^2+^ is a highly reactive ion, showing pro-oxidative properties which can lead to the oxidation and discoloration of meat products. Iron ion in ferrous state is more reactive than in the ferric state, however, ion in Fe^3+^ form shows pro-oxidant activity in presence of strong reducing agent [51]. Moreover, meat has its own oxidation preventing compounds, e.g., superoxide dismutase glutathione peroxidase. Thermal process could interfere with meat antioxidant systems, while mild heat treatment may change the protein structure and, therefore, increase the antioxidant capacity. On the other hand, the increase of temperature may cause reverse effect via meat compounds thermooxidation [47]. As it was mentioned earlier, despite the fact that ABTS^·+^ and DPPH methods allow to define free radical inhibition properties, the specificity of those methods may be responsible for the various results [48,49]. However, despite good results presented in nitrite-free sample it did not translate into the preservation of the color of the product (Table 2) or the inhibition of secondary lipid oxidation products formation (Table 4).

### 3.3. Secondary Lipid Oxidation Products and pH Values Analysis

The canned pork could be a source of various biological (bacteria: *L. monocytogenes, C. botulinum, Salmonella* sp.), chemical (nitrite, nitrate, N-nitrosamines, secondary lipid oxidation products), and physical (bones and their fragments, bristles) components (Table 1). The chemical hazard markers were presented in Table 4 and Table 5 and the microbiological ones in Table 6.

At the beginning of storage, no significant (*p* < 0.001) differences in TBARS levels were noted between samples. During storage, the amount of secondary lipid oxidation products increased in all the samples; however, a sharp change was observed only in the nitrite-free sample (from 0.006 mg MDA/kg at day 1 to 0.396 mg MDA/kg at day 60 indicating a difference of about 0.39 mg MDA/kg; NH sample—difference of about 0.032 mg MDA/kg; N sample—difference of about 0.021 mg MDA/kg). It was noted that during the storage period, the sample with 100 mg/kg of nitrite addition was more stable than the sample with the reduced amount of nitrite (50 mg/kg). However, at the end of the storage period, meat products containing various amounts of nitrite presented the same amount of TBARS.

Eskandari et al. [45] noted that the addition of 40 and 120 mg/kg of sodium (III) nitrite to frankfurters contributed to the effectively lowered amount of malondialdehyde during 8 weeks of storage (4 °C). Jin et al. [54] also noted strong antioxidant properties in pork sausages. In their experiment author decides to compare a product with nitrite addition to products with vegetable and/or fruit powder addition, which are good source of polyphenol compounds, and are also nitrate source. After 4 weeks of storage the value measured by TBARS method was found lower in samples with nitrite addition and with celery powder (no-nitrite) addition (probably thanks to the presence of polyphenol compounds). Similarly, Nowak et al. [55] observed that the addition of 18,000 mg/kg of curing salt (contains 0.5% of NaNO_2_ and 99.5% of salt) was sufficient to inhibit lipid oxidation in pork sausage during 28 days of storage in a greater case than sample with only salt addition (18,000 mg/kg). Low concentration of nitrite (20 mg/kg) caused significant (*p* < 0·001) inhibition of lipid oxidation, measured by the TBA test, in a cooked muscle system and 50 mg/kg nitrite concentration resulted in a highly significant (*p* < 0·001) reduction in TBA values [56].

Nitrite protects meat products from developing a rancid flavor. In general, for retarding rancidity and achieving the characteristic flavor in cured meat products, it was shown that the addition of 20–50 mg/kg of nitrite is required [4]. This was also confirmed by the results of sensory analysis (Table 7). As expected, the highest palpable rancid taste and aroma were found in nitrite-free sample—both directly after production (1.87 and 1.09 c.u.) and after six months (2.98 c.u. for both the parameters) of storage. However, an increase of rancidity taste and aroma was also observed after 180 days of storage in samples with nitrite addition. Moreover, in those samples, a sharp increase in the values of the parameters was observed—about 1.56 c.u. in NH sample and 1.12 c.u. in N sample for taste and about 1.75 c.u. and 1.34 c.u. for aroma, respectively. However, it should be kept in mind that nitrite can also exert pro-oxidative effects and cause oxidation of both lipids and proteins. This negative impact relates to the formation of peroxynitrite, a strong peroxide, from nitric oxide. Thus, nitrite can not only act as an antioxidant but also as a pro-oxidant [55].

The pH values were not stable during the 180 days of storage, and alternating declines were noticed throughout this period (Table 4). Similar results were observed by Choi et al. [57] and Shin et al. [58], whereby the pH of meat products generally decreased during storage. Rubio et al., 2007 pointed out that the major factor influencing the decrease in the pH was dissolution of CO_2_ into the pork patties during storage. The pH values of canned pork were significantly higher in NH containing 50 mg/kg of sodium nitrite than in the other treatments. The higher pH values may be attributed to the production of ammonia and other basic substances caused by proteolysis resulting from the growth of microorganisms during manufacturing process [59].

### 3.4. Nitrosamine, Cholesterol, Nitrite and Nitrate Residues and Water Activity (a_w_) Analysis

After 1st (day zero) and 180 days of storage, the canned pork products were tested for the presence of the following N-nitrosamines: N-nitrosodibutylamine, N-nitrosodiethylamine (NDEA), N-nitrosodimethylamine (NDMA), N-nitrosodipropylamine (NDPA), N-nitrosomorpholine, N-nitrosopiperidine (NPIP), and N-nitrosopyrrolidine (NPYR) (results expressed as μg/kg). Nitrite and nitrate residues were also measured. Results are presented in Table 5. At the beginning of the storage the highest amount of cholesterol was noted for the sample with 100 mg/kg nitrite addition (673 mg/kg). Nitrite free sample and sample with sodium nitrite addition reduced by half presented similar values (about 621 mg/kg). After 180 days of storage, cholesterol content decreased in NF sample (about 82 mg/kg) and increase in sample NH (about 50 mg/kg). Sample N seems to be constant.

It was found that, directly after production, NDPA was present in the sample with the maximum amount of sodium (III) nitrite (at a level of 0.6 μg/kg). In addition, directly after production, the nitrate content was 15 and 13 mg/kg in the N and NH samples, respectively, while nitrite content was below the limit of quantification—similar results were observed for the content of nitrite and nitrate in the NF samples. Considering the consumption safety of the products, the hazards and elimination methods for canned meat, summary based on our research and literature data, are presented in Table 1. According to Table 1, the acceptable quantity of selected N-nitrosamines (NDMA, NDEA, NPYR, NPIP) was set at <10 μg/kg. In our study, only one N-nitrosamine compound was detected but its level was far below of limit of quantification.

Heating could increase the COPs amount through the decrease of the antioxidant enzyme activity in denatured proteins [60]. The authors noted that temperature affected the decrease of cholesterol content through the conversion of this molecule into the 7-ketocholesterol (indicator of food oxidation). Min et al. [60] pointed out that, in general, various thermal treatment causes decrease of moisture and increases the levels of cholesterol and fat. Therefore, the decreased level of cholesterol can be connected with partial oxidation during thermal treatment and storage. This could be an explanation for the decrease of cholesterol content and increase of the TBARS parameter in sample NF.

In the case of nitrite content, some authors [45,61,62] pointed out that nitrite is a reactive substance and presents the ability to be bound with other substances in meat, and therefore indicates the tendency to disappear in meat products. Nitrite is sensitive to not only various processes during meat product production (storage conditions, thermal treatment) but also to other ingredients and meat composition [45,61,62]. Therefore, the differences between initial amount of nitrite versus the amount in the product can occur [45,61,62]. In our study, the nitrite residues have been detected only in sample with 100 mg/kg of sodium (III) nitrite addition (Table 5). The decrease in residual nitrite in N, NH, and NF were 10 mg/kg, <10 mg/kg, and <10 mg/kg, respectively. Similar results were noted by Eskandari et al. [40] in frankfurters in which no nitrite has been found after 1st day of storage (sample with 40 mg/kg of nitrite addition). These results agree also with those of Sindelar et al. (2007) [44], who concluded that the residual nitrite level decreases over time while in storage. In terms of health, when the content of residual nitrite is low, it is considered that the production of nitrosamine is low in meat products. Drabik-Markiewicz et al. [63] found that the occurrence of nitrosamine is highly related to the high temperature (> 200 °C) of meat products rather than to nitrite content.

However, it was observed that nitrite addition more significantly affected water activity (a_w_), and the difference was clearly dependent on the amount of the additive. At the beginning of the storage no significant (*p* < 0.001) differences have been noted between samples (Table 5). After six months of storage a sharp decrease in value of a_w_ parameter has been noted—decrease was about 0.009 for NF sample, and 0.015 and 0.022 units for NH and N samples, respectively.

To summarize it should be mentioned that according to the safety-related factors of the canned products (Table 1), after 180 days of storage, canned pork with reduced amount of nitrite did not exceed the safety indicator associated with lipid oxidation (N-nitrosamine presence, TBARS value). This suggests that the nitrite-free sample may pose a risk to consumer safety, as a high malondialdehyde amount indicates the ongoing oxidation processes, and also in the sample with 100 mg/kg addition of nitrite the risk of nitrosamine formations.

### 3.5. Microbiological Analysis 

The most effective method for controlling the growth of spores and vegetative cells is the addition of sodium nitrite.

The ability of this salt to inhibit bacterial species could be related to the formation of nitrous acid or nitrous oxide (NO), and subsequently, peroxynitrite [64,65,66]. Chang et al. [64] stated that NO has the ability to inhibit the synthesis of adenosine triphosphate through the destruction of iron-sulfur enzymes in bacterial cells. In addition, Majou and Christieans [66] highlighted that anaerobic conditions are helpful for the formation of peroxynitrite precursors. The results of the experiments conducted in the present study (Table 6) showed that selected bacterial species (*C. perfringens, L. monocytogenes, Escherichia coli,* and *Salmonella)* were not detected in the samples with nitrite addition as well as nitrite-free samples. This may indicate that good hygiene practice was maintained during production, the raw meat material was not infected, and the appropriate sterilization process was carried out. Moreover, Jackson et al. [61] mentioned that 0.2 mg of sodium nitrite per mL was sufficient to inhibit *C. perfringens* at 20 °C. According to the U.S law the amount of nitrite added to meat products is limited to 120 mg/kg—bacon, 156 mg/kg—frankfurters, 200 mg/kg—ham. (120, Authors noted that the addition of powders or juices, mostly from celery, did not prevent from bacteria growth in “natural-cured” product during 10 days of storage. In case of non-compliance with the temperature for meat products or in case of microbiological contamination of “natural-cured” or uncured products they are unprotected. Also, Sullivan et al. [67] noted that the addition of only preconverted celery juice powder to uncured, inoculated with *L. monocytogenes* strains ham did not cause inhibition properties during 35 days of storage. Only the additional addition of natural antimicrobial and nitrate reducing starter culture caused similar inhibition process as the one in conventionally cured ham. Sindelar and Milkowski [44] pointed out that nitrite is well-known to suppress the outgrowth of *C. botulinum* spores in cured meat products and controlling the growth of other pathogens, such as *Listeria monocytogenes, Bacillus cereus, Staphylococcus aureus,* and *Clostridium perfringens* [3].

### 3.6. Sensory Analysis

The results of the sensory evaluation of canned pork, including odor, flavor, color, and overall quality, are summarized in Table 7. The addition of a reduced amount of sodium (III) nitrite (50 mg/kg) to canned meat gave similar results obtained with the addition of 100 mg/kg of this additive. Significant differences were noted in the color of meat after 1st and 180 days of storage (Table 7). A significantly lower (*p* < 0.001 *) proportion of redness was observed by the evaluators on the cross-section of the NF sample compared to the N and NH samples. A higher intensity of rancid aroma (*p* < 0.001 *) and green color (*p* = 0.005 *) was observed in the sample NF compared to the N and NH samples after 1st and N sample after 180 days of storage, respectively. In addition, after 180 days of storage, the hardness of samples increased significantly in the following order: NF<NH<N. Significantly lower (*p* < 0.036 *) compactness of the product was observed in the NF sample compared to the N and NH samples. The overall quality of the samples NH and N was rated as 5.69 c.u. Moreover, various interconnected biochemical processes such as Maillard and oxidative reactions, thiamine degradation etc. also contribute to create aroma. The oxidation of fat leads to the creation of volatile esters, acids, aldehydes, etc., which may not only be responsible for cured aroma, but also for rancidity. Lack of presence of nitrite in meat products can cause high aldehyde generations and reduce the odor of sulfur compounds which are found as main compounds of cooked meat. However, the absence or presence of nitrite shows no effect on sulfur compounds itself [68].

In contrast to odor, flavor of cured meat products is the least understood factor. Alahakoon [3] pointed out that sodium nitrite in the amount of 50 mg/kg allows to create noticeable, typical cured flavor. 

As it was mentioned earlier MDA measurement is one of the oxidation process indicators and the presence of this aldehyde, even in low amounts, causes rancid aroma in product. Following Domínguez et al. [69], the acceptable limits for MDA in meat products which does not create negative odor is 2–2.5 mg/kg. According to our data (Table 1) the limit for MDA amount has been set below 2 mg/kg. Although no-nitrite sample exceeded this value (Table 4), it was rated definitely worse compared to other samples. Our observation confirmed that if nitrite is not added, canned meat will be considered as an unappealing product (overall quality: 3.91 c.u.).

### 3.7. Verification of the Differences between Canned Pork with Various Amounts of Nitrite Using Principal Component Analysis

To illustrate the differences between canned pork with various amounts of nitrite, PCA was applied. The first two main components obtained in the analysis explained about 64.41% of the total variance (first component: 37.81%; second component: 26.6%) (Figure 2). The first component (PC1) indicated a high content of nitrosylhemochrome, high a* value, highly intense color of meat and fat, high compactness, and high overall quality as well as a less intense rancid aroma, metallic aroma, and rancid taste, low L* and b* values, low level of TBARS, and a less intense green color in the product. The second component (PC2) indicated a high hardness, high ABTS˙^+^ reducing capacity, highly intense metallic taste, and high nitrosylhemochrome concentration, while a low content of FRAP, less juiciness, low pH, low water activity.

The individual factor map (Figure 3) presented the differences of samples with various amount of nitrite addition. Figure 3 showed that NF samples were quite different from the N and NH samples. NF samples showed lower (at T = 0) or very lower (at T= 180) values of the first component (PC1), while N and NH samples showed high or slightly low values of the first component (PC1). Therefore, second component seems connected with NF sample, where samples N and NH seem to be more connected with the first component. Moreover, the N and NH samples were quite mixed up. However, at T = 0, they had low values of the second component (PC2), while at T = 180 they had high values. Samples N and NH presented strongly positive correlations, whereas samples NF strongly negative correlations with the analyzed components. The contrast between sample NF and samples N and NF is very strong.

## 4. Conclusions

The canned pork with reduced amount of sodium nitrite could be a source of various biological (bacteria: *L. monocytogenes*, *C. botulinum*, *Salmonella* sp.), chemical (nitrite, nitrate, N-nitrosamines, secondary lipid oxidation products), and physical (bones and their fragments, bristles etc.) hazards. Fortunately, some of these health hazards can be limited by complying with the principles of good hygiene practice (e.g., physical, biological, and chemical risks) and properly carrying out heat treatment. Based on the obtained research results, the following detailed conclusions were formulated first it is not possible to completely eliminate nitrite from the canned pork production process without deteriorating the color, antioxidant properties, sensory characteristics, and health safety. However, it has been noted that the level of chemicals, such as N-nitrosamines, nitrate, and nitrite residues, can be limited by decreasing the amount of nitrite addition to 50 mg/kg. Reducing the addition of nitrite in the production of canned pork to 50 mg/kg resulted in the formation of a pinkish-red color of cured meat. However, this color was unstable and faded during 180 days of storage. Also, the free-radical scavenging ability for sample with 50 mg/kg of sodium nitrite was observed to be poor, so fortification with plant material rich in various polyphenolic substances may be necessary.

## Figures and Tables

**Figure 1 foods-09-01869-f001:**
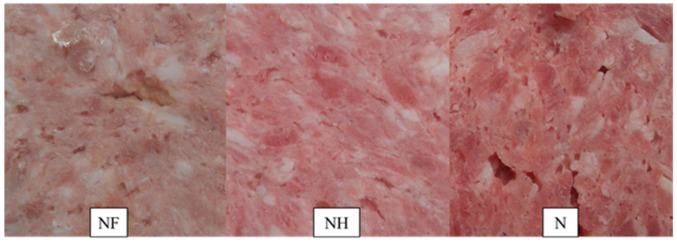
Visual aspects of canned meat with various amounts of nitrite after 180 days of storage. NF—nitrite-free, NH—50 mg kg^−1^ of nitrite addition, N—100 mg kg^−1^ of nitrite addition.

**Figure 2 foods-09-01869-f002:**
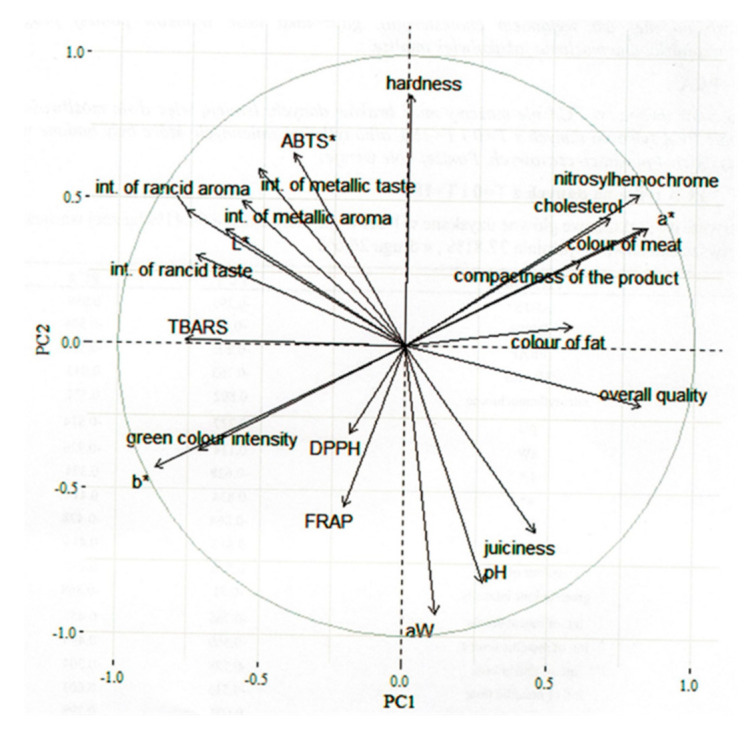
The variables factor map (PCA).

**Figure 3 foods-09-01869-f003:**
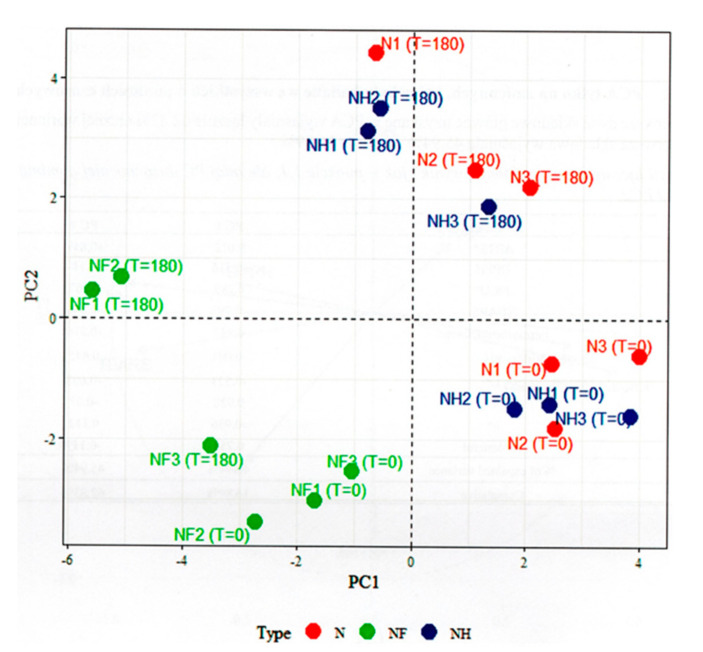
The individuals factor map (PCA).

**Table 1 foods-09-01869-t001:** Main microbiological and chemical substances of safety concern, their mechanism of generation, and health risks.

Category of Hazard	Type of Hazard	Limits	Risks for Health	Preventive Measures	References
MICROBIOLOGICAL	*L. monocytogenes*	Absence in 25 g Before the food has left the immediate control of the food business operator, who has produced it or <100 cfu/g Products placed on the market during their shelf-life	At a dose above 1000 cfu/g may cause listeriosis	pH < 5.0 and a_w_ < 0.94 or pH < 4.4 and a_w_ < 0.92	[21]
	*C. botulinum*	Absence in 1 g	Botulinum neurotoxin intoxication (MID = 5–10 ng botulinum toxin)	pH 4.8–8.0 a_w_ < 0.94 Sterilization process: 121 °C, 5 min	[22]


	*Salmonella* sp.	Absence in 25 g	At a dose above 10^5^ cfu/g may cause salmonellosis	pH 4–8 a_w_ < 0.92 Pasteurization process	[21]
CHEMICAL	Nitrate (E251-252)	Max. 100 mg/kg	May cause methemoglobinemia	Dose control	[10]
	Nitrite (E249-250)	Max. 100 mg/kg			
	Secondary lipid oxidation products, oxidized fatty acids, oxylipins, cholesterol oxides	TBARS <2.0 mg/kg and organoleptic analyses of intensity of the rancid taste/rancid aroma indicator <3 c.u.	Compounds initiating carcinogenesis in humans Lower nutritive value and poor sensory quality Mutagenicity, cytotoxicity	Addition of antioxidant Proper closing of products	[23,24,25]
	N-nitrosamines (NDMA—N-nitrosodimethylamine, NDEA—N-nitrosodiethylamine, NPIP—N-nitrosopiperidine, NPYR—N-nitrosopyrrolidine)	<10 μg/kg Total volatile nitrosamine content for meat products No maximum permitted limits have been set for nitrosamines in food products in EU countries 10 μg/kg is the US limit for meat products	Compounds initiating carcinogenesis in humans	Reduction of nitrite addition Addition of reduction substances (ascorbate, erythorbate, tocopherol)	[26]

**Table 2 foods-09-01869-t002:** Changes in color properties of canned meat with reduced amount of nitrite during storage.

Parameter	Sample	Storage Time (days)			
		1	60	90	180
Nitrosylhemochrome [mg/kg]	NF	1.28 ± 0.23 ^Cb^	1.16 ± 0.21 ^Cb^	2.25 ± 0.28 ^Cab^	3.31 ± 0.74 ^Ca^
	NH	26.52 ± 0.65 ^Bb^	22.19 ± 1.23 ^Bb^	24.02 ± 1.27 ^Bb^	18.37 ± 1.11 ^Ba^
	N	32.27 ± 0.51 ^Aa^	28.16 ± 1.16 ^Ab^	30.12 ± 1.67 ^Aab^	31.50 ± 1.15 ^Aab^
	N	61.37 ± 0.44 ^Bb^	61.86 ± 0.43 ^Ab^	62.95 ± 0.34 ^Bab^	64.67 ± 0.60 ^Aa^
L*	NH	62.28 ± 0.55 ^Bb^	61.51 ± 0.85 ^Ab^	62.92 ± 0.68 ^Bab^	64.83 ± 0.49 ^Aa^
	NF	63.72 ± 0.60 ^Abc^	63.28 ± 0.52 ^Ac^	65.73 ± 0.52 ^Ab^	66.69 ± 0.86 ^Aa^
	N	11.29 ± 0.18 ^Aab^	11.02 ± 0.30 ^Aab^	11.55 ± 0.22 ^Aa^	10.49 ± 0.17 ^Ab^
a*	NH	10.81 ± 0.27 ^Aa^	9.90 ± 0.45 ^Aa^	11.06 ±0.28 ^Aa^	9.89 ± 0.18 ^Ba^
	NF	5.86 ± 0.26 ^Ba^	5.11 ± 0.15 ^Bb^	6.16 ± 0.17 ^Ba^	6.40 ± 0.10 ^Ca^
	N	8.60 ± 0.17 ^Ba^	8.95 ± 0.30^Ba^	9.19 ± 0.11 ^Ba^	8.54 ± 0.18 ^Ba^
b*	NH	8.27 ± 0.13 ^Bb^	9.48 ± 0.28 ^Ba^	8.96 ± 0.19 ^Bab^	8.64 ± 0.17 ^Bb^
	NF	11.92 ± 0.26 ^Aa^	12.47 ± 0.18 ^Aa^	12.28 ± 0.17 ^Aa^	12.21 ± 0.17 ^Aa^

NF—nitrite-free, NH—50 mg kg^−1^ of nitrite addition, N—100 mg kg^−1^of nitrite addition. Means with different capital letters are significantly different (*p* < 0.05) in the same column. Means with different small letters are significantly different (*p* < 0.05) in the same row. Values are presented as mean ± standard error (SE).

**Table 3 foods-09-01869-t003:** Changes in antioxidant properties of canned meat with reduced amount of nitrite during storage.

Parameter	Sample	Storage Time (days)			
		1	60	90	180
ABTS [mg/mL]	NF	4.79 ± 0.02 ^Ba^	2.55 ± 0.10 ^Bb^	3.06 ± 0.08 ^Bc^	2.93 ± 0.08 ^Ab^
	NH	5.09 ± 0.18 ^Ba^	3.62 ± 0.22 ^Abc^	3.78 ± 0.09 ^Ab^	2.83 ± 0.34 ^ABc^
	N	5.62 ± 0.16 ^Aa^	3.61 ± 0.06 ^Ab^	4.05 ± 0.20 ^Ab^	2.12 ± 0.12 ^Bc^
DPPH [mg/mL]	NF	0.01 ± 0.00 ^Bc^	0.02 ± 0.00 ^Aa^	0.02 ± 0.00 ^Ab^	0.02 ± 0.00 ^Aa^
	NH	0.02 ±0.00 ^Aa^	0.02 ± 0.00 ^Aa^	0.01 ± 0.00 ^Bb^	0.01 ± 0.00 ^Bb^
	N	0.02 ± 0.00 ^Aa^	0.02 ± 0.00 ^Ba^	0.02 ± 0.00 ^Aa^	0.02 ± 0.00 ^Ba^
FRAP [A_700_]	NF	2.10 ± 0.13 ^Aa^	1.82 ± 0.04^Ab^	1.19 ± 0.02 ^Bc^	1.83 ± 0.03 ^Ab^
	NH	1.50 ± 0.08 ^Ba^	1.67 ± 0.04 ^Aa^	1.53 ± 0.04 ^Aa^	1.63 ± 0.03 ^Ba^
	N	1.99 ± 0.04 ^Aa^	1.36 ± 0.04 ^Bc^	1.10 ± 0.04 ^Bd^	1.59 ± 0.08 ^Bb^

NF—nitrite-free, NH—50 mg kg^−1^ of nitrite addition, N—100 mg kg^−1^of nitrite addition. Means with different capital letters are significantly different (*p* < 0.05) in the same column. Means with different small letters are significantly different (*p* < 0.05) in the same row. Values are presented as mean ± standard error (SE).

**Table 4 foods-09-01869-t004:** Changes in lipid oxidation and pH values of canned meat with reduced amount of nitrite during storage.

Parameter	Sample	Storage Time (Days)			
		1	60	90	180
TBARS [mg MDA/kg]	NF	0.01 ± 0.07 ^Ac^	0.40 ± 0.02 ^Aa^	0.31 ± 0.02 ^Ab^	0.35 ± 0.03 ^Ab^
	NH	0.01 ±0.48 ^Ab^	0.04 ± 0.00 ^Ba^	0.02 ± 0.00 ^Bb^	0.02 ± 0.00 ^Bb^
	N	0.01 ± 0.11 ^Ab^	0.03 ± 0.00 ^Ba^	0.02 ± 0.01 ^Ba^	0.02 ± 0.00 ^Ba^
pH	NF	6.68 ± 0.00 ^Ba^	6.51 ± 0.01 ^Ab^	6.30 ± 0.01 ^Ac^	6.31 ± 0.02 ^Ac^
	NH	6.73 ± 0.00 ^Aa^	6.62 ± 0.01 ^Ab^	6.44 ± 0.01 ^Ac^	6.26 ± 0.04 ^Ad^
	N	6.68 ± 0.00 ^Ba^	6.61 ± 0.01 ^Aab^	6.20 ± 0.22 ^Ab^	6.20 ± 0.05 ^Ab^

NF—nitrite-free, NH—50 mg kg^−1^ of nitrite addition, N—100 mg kg^−1^ of nitrite addition. Means with different capital letters are significantly different (*p* < 0.05) in the same column. Means with different small letters are significantly different (*p* < 0.05) in the same row. Values are reported as mean ± standard error.

**Table 5 foods-09-01869-t005:** Nitrosamine, cholesterol, nitrite and nitrate residues and water activity (a_w_) amounts of canned pork with reduced amount of nitrite addition during storage.

Parameter		Storage Time (Days)				
		1		180		
	Sample					
	NF	NH	N	NF	NH	N
N-Nitrosodibutylamin (NDBA) [µg/kg]	<0.5	<0.5	<0.5	<0.5	<0.5	<0.5
N-Nitrosodiethylamin (NDEA) [µg/kg]	<0.5	<0.5	<0.5	<0.5	<0.5	<0.5
N-Nitrosodimethylamin (NDMA) [µg/kg]	<0.5	<0.5	<0.5	<0.5	<0.5	<0.5
N-Nitrosodipropylamin (NDPA) [µg/kg]	<0.5	<0.5	0.6	<0.5	<0.5	<0.5
N-Nitrosomorpholin (NMOR) [µg/kg]	<0.5	<0.5	<0.5	<0.5	<0.5	<0.5
N-Nitrosopiperidin (NPIP) [µg/kg]	<0.5	<0.5	<0.5	<0.5	<0.5	<0.5
N-Nitrosopyrrolidin [µg/kg]	<0.5	<0.5	<0.5	<<0.5	<0.5	<0.5
Cholesterol [mg/kg]	622 ± 124	620 ± 124 ^Cb^	673 ± 135 ^Aa^	560 ± 110 ^Cb^	670 ± 130 ^Ba^	673 ± 135 ^Aa^
Nitrate as NaNO_3_[mg/kg]	<10 ^Ca^	13 ± 3 ^Ba^	15 ± 3 ^Aa^	<10 ^Ba^	<10 ^Bb^	15^Ab^
Nitrite as NaNO_2_[mg/kg]	<10 ^Ba^	<10 ^Ba^	10 ± 2 ^Ab^	<10 ^Ba^	<10 ^Ba^	10 ^Aa^
a_w_	0.984 ± 0.00 ^Aa^	0.982 ± 0.00 ^Aa^	0.982 ± 0.00 ^Aa^	0.975 ± 0.00 ^Ab^	0.967 ± 0.00 ^Bb^	0.960 ± 0.00 ^Cb^

NF—nitrite-free, NH—50 mg kg^−1^ of nitrite addition, N—100 mg kg^−1^ of nitrite addition. Means with different capital letters are significantly different (*p* < 0.05) in the same column. Means with different small letters are significantly different (*p* < 0.05) in the same row. Values are reported as mean ± standard error (SE).

**Table 6 foods-09-01869-t006:** Microbiological analysis of canned pork with reduced amount of nitrite during storage.

Parameter	Time (days)	Sample (Mean ± SE)		
		NF	NH	N
*C. botulinum* [cfu/g]	1	<10	<10	<10
	180	<10	<10	<10
*L. monocytogenes* [cfu/g]	1	<10	<10	<10
	180	<10	<10	<10
*Salomonella* [in 25 g]	1	n.d.	n.d.	n.d.
	180	n.d.	n.d.	n.d.
TVC [cfu/g]	1	<10	<10	<10
	180	<10	<10	<10

NF—nitrite-free, NH—50 mg kg^−1^ of nitrite addition, N—100 mg kg^−1^ of nitrite addition. <—the parameter result is below the limit of quantification, n.d.—not detected.

**Table 7 foods-09-01869-t007:** Sensory analysis of canned pork with reduced amount of nitrite during storage.

Parameter	Time (Days)	Sample (Mean ± SE)			Linear Mixed Models (p)	
		NF	NH	N	NF vs N	NH vs N
Meat color (c.u.)	1	2.56 ± 1.65	8.83 ± 0.83	8.6 ± 1.07	*p* < 0.001 *	*p* = 0.556
	180	3.32 ± 1.79	7.2 ± 1.43	8.4 ± 1.08	*P* < 0.001 *	*p* = 0.03 *
Fat color (c.u.)	1	6.93 ± 2.68	8.74 ± 1.49	8.57 ± 1.62	*p* = 0.062	*p* = 0.773
	180	7.58 ± 1.88	7.97 ± 2.07	8.03 ± 1.94	*p* = 0.549	*p* = 0.912
Green color intensity (c.u.)	1	1.36 ± 1.61	0.26 ± 0.33	0.23 ± 0.28	*p* = 0.024 *	*p* = 0.801
	180	1.00 ± 0.90	0.46 ± 0.81	0.17 ± 0.19	*p* = 0.005 *	*p* = 0.183
Compactness (c.u.)	1	7.37 ± 2.35	7.66 ± 1.88	8.27 ± 1.55	*p* = 0.422	*p* = 0.609
	180	6.67 ± 2.03	8.32 ± 1.54	8.32 ± 1.54	*p* = 0.036 *	*p* = 1
Juiciness (c.u.)	1	7.34 ± 2.04	7.86 ± 1.54	7.69 ± 1.52	*p* = 0.705	*p* = 0.848
	180	4.28 ± 3.10	4.14 ± 3.60	4.14 ± 3.60	*p* = 0.778	*p* = 1
Hardness (c.u.)	1	4.98 ± 1.95	4.99 ± 1.90	5.51 ± 2.01	*P* = 0.584	*p* = 0.601
	180	5.91 ± 2.21	7.87 ± 1.34	7.87 ± 1.34	*p* = 0.01 *	*p* = 1
Overall quality (c.u.)	1	5.51 ± 2.19	8.28 ± 1.94	8.62 ± 1.64	*p* = 0.005*	*p* = 0.795
	180	3.91 ± 2.23	5.69 ± 3.54	5.69 ± 3.54	*p* = 0.146	*p* = 1
Rancid odor (c.u.)	1	1.09 ± 0.74	0.29 ± 0.28	0.22 ± 0.19	*p* < 0.001 *	*p* = 0.437
	180	2.98 ± 2.68	2.04 ± 1.84	1.56 ± 2.18	*p* = 0.187	*p* = 0.612
Metallic odor(c.u.)	1	0.88 ± 1.03	0.32 ± 0.38	0.41 ± 0.45	*p* = 0.133	*p* = 0.572
	180	1.17 ± 1.28	1.49 ± 1.42	0.96 ± 1.11	*p* = 0.653	*p* = 0.331
Rancid flavor (c.u.)	1	1.87 ± 2.23	0.48 ± 0.40	0.44 ± 0.38	*p* = 0.032 *	*p* = 0.758
	180	2.98 ± 2.68	2.04 ± 1.84	1.56 ± 2.18	*p* = 0.187	*p* = 0.612
Metallic flavor (c.u.)	1	0.66 ± 0.87	0.37 ± 0.41	0.43 ± 0.50	*p* = 0.264	*p* = 0.97
	180	1.17 ± 1.28	1.49 ± 1.42	0.96 ± 1.11	*p* = 0.653	*p* = 0.331

NF—nitrite-free, NH—50 mg kg^−1^ of nitrite addition, N—100 mg kg^−1^of nitrite addition. Values are presented as mean ± standard error (SE). * statistically significant (*p* < 0.05).

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
