# Peer review of "The Possibility of Reduction of Synthetic Preservative E 250 in Canned Pork"

_foods, 2020, doi:10.3390/foods9121869_

Round 1

Reviewer 1 Report

The present paper shows the results of three variants of canned pork by adding different amounts of sodium nitrite with  the objective of determining the lowest possible amount of nitrite that can be added to canned meat in comparison to the recommended levels. In my opinion, it is a well strcutured and very interesting work. The Introduction is well documented and Material and Methods are described in detail. Results are properly written with a large range of data and well discussed facts.
Minor points:
I would suggest writing the objective of the work in the abstract.

The present paper shows the results of three variants of canned pork by adding different amounts of sodium nitrite with  the objective of determining the lowest possible amount of nitrite that can be added to canned meat in comparison to the recommended levels. In my opinion, it is a well structured and very interesting work. Results are properly written with a large range of data and well discussed facts. Thus, in my opinion, it merits to be published in Foods.
Minor points:
Abstract:
I would suggest writing the objective of the work in the abstract.

Introduction:
The introduction is well documented, but it would be useful to have more updated references.
Material and methods:
Please, describe some points of the methodology in more detail:
L 83: Water: distillate water?
Please specify the producers of the reactive and standards (L120: ABTS, DPPH; L 129: trolox etc)
Results:
Results are properly written with a large range of data. However, Figure 3, should be more commented trying to explain the situation of the samples in the individual factor map

Author Response

Review Report 1 (Round 1)

Abstract:
I would suggest writing the objective of the work in the abstract.

Thank you for this remark. Objective of the work has been added to the abstract according with reviewer suggestion. Lines in text: 11-12

Introduction:
The introduction is well documented, but it would be useful to have more updated references.
Thank you for this remark. In the introduction, the authors added articles from the last five years. Lines in text: 60-64.

Material and methods:
Please, describe some points of the methodology in more detail:
L 83: Water: distillate water?

Thank you for this remark. Technological water was used in production. Line in text: 97

Please specify the producers of the reactive and standards (L120: ABTS, DPPH; L 129: trolox etc)
Thank you for this remark. Missing information has been added. Lines in text: 124, 137, 146, 153, 154, 155

Results:
Results are properly written with a large range of data. However, Figure 3, should be more commented trying to explain the situation of the samples in the individual factor map

Thank you for this valuable tip. The authors extended the discussion of the results related to Figure 3. Lines in text: 588-592

Reviewer 2 Report

The paper proposes the study of the different hazards that could be observed in a cooked meat product like canned pork, by reducing the nitrite level at half and total dose.

Some considerations and modifications should be carried out in the manuscript to improve the present format.

Table 1 should be corrected in a horizontal orientation and physical hazards should be deleted from the table since only biological and chemical hazards are considered in this work.

In the material and methods, why is the cholesterol included in this section? I think that it would be interesting to reorganized this section, since quality characteristics of the products are described, apart from their implications on safety. So, may be two sections by indicating Canned pork preparation and Quality atributtes/characteristics could be established.

In the sensory analysis section, which type of cards were used? Were included some pictures of the products on them or were only a text card? Please more details in this sense.

L216: what do authors mean with a unique simple number?

L277: Some others color parameters such as Chroma and Hue angle should have been considered since the “tone” is considered in the text as part from the discussion.

In general, in all Tables, must be indicated the same decimals quantity as the value number, in all rows/columns.

L330-332: could authors indicate in the text some data to support this confirmation?

L344-362: Reduce this paragrah.

L379-384: Delete this paragrah

In all text, please indicate as units ppm or mg/Kg, but not both.

L407: Delete this stament.

In Table 5, could authors explain more extensively the reason because of varying the cholesterol level by effect of the time?

L455-461: Delete this text.

L485-488: please, clarify this lines.

L491-495: Delete this text.

L510-511 can be joined as following: Jackson et al (60) mentioned that …..at 20ºC in meat products ….no nitrite addition.

Thus delete Authors decide to inoculate commercially available

L517-519: This explanation is, from my point of view, is unnecessary, please remove it.

L540-543: Delete this paragrah.

L548-550: Rewritte this sentence.

Please in Table 7 change the atributtes denomination:

Meat color

Fat color

Compactness

Rancid odor

Metallic odor

Rancid Flavor

The only real tastes recognized are the following four: sweet, salted, bitter and acid. The other components are flavors of the products. The word aroma should be substitute by odor since this is the sensorial perception perceived by nose while aroma is related with any organic substance present in the product.

Reduce the number of references

Conclusions should be integrated all together in a paragrah by joining the whole knowledge of the experimental design.

Author Response

Review Report 2 (Round 1)

Table 1 should be corrected in a horizontal orientation and physical hazards should be deleted from the table since only biological and chemical hazards are considered in this work.

Thank you for valuable tip. The physical hazards have been delated.

Physical hazards were included in the Table 1 to present all potential hazards which are expected in a canned meat (according to the HACCP rules: biological/microbiological, chemical and physical hazards). We agree with the opinion of the reviewer. In Table 1, we leave only microbiological and chemical hazards that may be affected by the concentration of sodium III nitrate used. Table 1 has been corrected in a horizontal orientation.

In the material and methods, why is the cholesterol included in this section? I think that it would be interesting to reorganized this section, since quality characteristics of the products are described, apart from their implications on safety. So, may be two sections by indicating Canned pork preparation and Quality atributtes/characteristics could be established.

Thank you for this remark. We agree with the opinion of the reviewer and for better clarification, section “Quality attributes” has been added. Line in text:121

In the sensory analysis section, which type of cards were used? Were included some pictures of the products on them or were only a text card? Please more details in this sense.

Thank you for this remark. Determination of the intensity of the features was carried out using an unstructured scale, anchored at an appropriate distance from the edges of the score card with words specifying the minimum and maximum intensity of the examined feature. A linear graphic scale (100-mm line which is converted into conventional units [c.u.]) was used for the assessment of sensory properties. An unstructured graphical scale was used for the sensory properties determination. Lines in text: 232-242

L216: what do authors mean with a unique simple number?

Thank you for this remark. For the sample description random numbers have been applied (eg. 243 for the one variant, 164 for the second variant ect.).

L277: Some others color parameters such as Chroma and Hue angle should have been considered since the “tone” is considered in the text as part from the discussion.

Thank you for valuable tip. Authors considered to focus on basic color parameters (lightness, redness and yellowness) because in context of canned meat the share of individual parameters in the overall tone of the color is more important than its intensity or hue angle.

In general, in all Tables, must be indicated the same decimals quantity as the value number, in all rows/columns.

Thank you for this remark. Data in tables have been corrected. However, various decimal quantity in some of the tables is connected to the importance of the correct interpretation of the results.

L330-332: could authors indicate in the text some data to support this confirmation?

Thank you for this remark. Missing data has been added. Lines in text: 348-351

L344-362: Reduce this paragrah.

Thank you for this remark. The paragraph has been reduced. Lines in text: 370-375

L379-384: Delete this paragrah

Thank you for this remark. The paragraph has been deleted. Lines in text: 398-403

In all text, please indicate as units ppm or mg/Kg, but not both.

Thank you for this remark. All units have been replaced by mg/kg.

L407: Delete this stament.

Thank you for this remark. The statement has been deleted. Line in text: 424-425

In Table 5, could authors explain more extensively the reason because of varying the cholesterol level by effect of the time?

Thank you for this remark. An explanation has been added. Lines in text: 476-478

L455-461: Delete this text.

Thank you for this remark. The text has been deleted. Lines from text: 472

L485-488: please, clarify this lines.

Thank you for this remark. Sentences have been rewritten for more clarification.  Lines: 500-505

L491-495: Delete this text.

Thank you for this remark. The text has been deleted. Lines in text: 507-510

L510-511 can be joined as following: Jackson et al (60) mentioned that …..at 20ºC in meat products ….no nitrite addition.

Thank you for this remark. The sentence has been changed. Lines in text: 520-526

Thus delete Authors decide to inoculate commercially available

Thank you for this remark. The text has been deleted. Lines in text: 527-529

L517-519: This explanation is, from my point of view, is unnecessary, please remove it.

Thank you for this remark. The text has been deleted. Lines in text: 532-534

L540-543: Delete this paragrah.

Thank you for this remark. The paragraph has been deleted. Lines in text: 557-559

L548-550: Rewritte this sentence.

Thank you for this remark. The sentence has been rewritten. Lines in test: 555-556

Please in Table 7 change the atributtes denomination:

Meat color

Fat color

Compactness

Rancid odor

Metallic odor

Rancid Flavor

The only real tastes recognized are the following four: sweet, salted, bitter and acid. The other components are flavors of the products. The word aroma should be substitute by odor since this is the sensorial perception perceived by nose while aroma is related with any organic substance present in the product.

Thank You for valuable tip. Attributes have been changed. Table 7.

Reduce the number of references

Thank you for this remark. All references have been carefully checked and some of them were removed.

Conclusions should be integrated all together in a paragrah by joining the whole knowledge of the experimental design.

Thank You for valuable tip. Conclusions have been integrated and organized. Lines in text: 600-607